# *Aurantiochytrium* sp. Meal Improved Body Fatty Acid Profile and Morphophysiology in Nile Tilapia Reared at Low Temperature

**Rosana Oliveira Batista** [1]**, Renata Oselame Nobrega** [1]**, Delano Dias Schleder** [2]**, James Eugene Pettigrew** [3] **and Débora Machado Fracalossi** [1,*]

[1] Departamento de Aquicultura, Universidade Federal de Santa Catarina (UFSC), Florianópolis 88034-001, SC, Brazil; rosana.engpesca@gmail.com (R.O.B.); renata.oselamenobrega@gmail.com (R.O.N.)

[2] Departamento de Medicina Veterinária, Instituto Federal Catarinense, Campus Araquari, Araquari 89245-000, SC, Brazil; delano.schleder@ifc.edu.br

[3] Pettigrew Research Services, Tubac, AZ 85646, USA; jim@pettigrewconsulting.com

\* Correspondence: debora.fracalossi@ufsc.br; Tel.: +55-48-3721-6300

**Abstract:** *Aurantiochytrium* sp. is a heterotrophic microorganism that produces docosahexaenoic acid (DHA), thus being considered as a possible replacement for fish oil in aquafeeds. We investigated the effect of *Aurantiochytrium* sp. meal (AM) dietary levels (0, 5, 10, 20, and 40 g kg$^{-1}$) on Nile tilapia body and hepatopancreas fatty acid (FA) profile, body FA retention, somatic indices, and morphophysiological changes in the intestine and hepatopancreas, after feeding Nile tilapia juveniles (average initial weight 8.47 g) for 87 days at 22 °C. The 10AM diet was compared to a control diet containing cod liver oil (CLO), since their DHA concentration was similar. Within fish fed diets containing increasing levels of AM, there was a linear increase in n-3 FA content, especially DHA, which varied in the body (0.02 to 0.41 g 100 g$^{-1}$) and hepatopancreas (0.15 to 1.05 g 100 g$^{-1}$). The morphology of the intestines and hepatopancreas was positively affected in AM-fed fish. Fish fed 10AM showed less accumulation of n-3 FAs in the body and hepatopancreas when compared to fish fed CLO. Therefore, AM is an adequate substitute for fish oil in winter diets for Nile tilapia, with the supplementation of 40AM promoting the best results regarding intestine and hepatopancreas morphophysiology.

**Keywords:** *Aurantiochytrium* sp.; docosahexaenoic acid; histology; *Oreochromis niloticus*; physiology; temperature

## 1. Introduction

Fish are ectothermic organisms; therefore, their body temperature is influenced by water temperature, which consequently affects their metabolic rate, feed consumption, feed conversion, and other physiological functions [1]. Water temperature is one of the most important abiotic factors in aquaculture, as it directly affects the growth and survival of fish [2]. Whereas in nature, fish may use behavioral responses to overcome low temperature, fish under farming conditions cannot use such natural responses. This is the case in pond or cage farming situations where fish may be subjected to extreme temperature fluctuations [3]. As a primary response to low temperature, fish increase serum cortisol and catecholamine levels; as a secondary response, metabolic changes may occur [3]. These adaptations are commonly used as indicators of short-term cold responses [4]. However, if low water temperatures persist for longer periods, the continued stress can trigger tertiary responses, causing physiological changes that can lead to mortality [5].

Nile tilapia, *Oreochromis niloticus*, is the third most cultivated fish species in the world [6]; despite its tropical origin, it has been introduced in many subtropical and temperate regions around the world. In Brazil, the fourth country in Nile tilapia production worldwide, the highest production occurs in the Southern states of Paraná, São Paulo, Minas Gerais, and Santa Catarina [7], where the climate is predominately subtropical, or altitude-tropical, with important thermal variations between summer and winter. Average water temperatures of 20–22 °C were reported in tilapia farms in Brazil during the winter [8,9]. The optimal temperature for Nile tilapia growth is around 26–30 °C [10]. Temperatures below 22 °C result in a drastic decrease in zootechnical performance [9,11]. Between 16 and 13 °C, feed intake ceases and, below 9 °C, voluntary movements cease [10].

The worldwide importance of Nile tilapia farming has led to an urgent need to develop new technologies to improve production in subtropical regions [12]. The availability of Nile tilapia strains that are more tolerant to low temperatures and the adoption of sustainable management practices can improve production in subtropical regions. However, research on winter diets for Nile tilapia, focusing on nutrients and biological responses, are also reported to improve performance [9,13–16].

Water temperature plays an important role in lipid metabolism in fish [17]. Polyunsaturated fatty acids (PUFA) of 18-carbon chains, such as $\alpha$-linolenic acid ($\alpha$-LNA,18:3 n-3) and/or linoleic acid (LOA, 18:2 n-6), meet the requirements of fatty acids for the optimal growth of Nile tilapia when reared in the ideal temperature range (26–30 °C). However, dietary supplementation with n-3 long-chain PUFA (n-3 LC-PUFA), such as docosahexaenoic acid (DHA, 22:6 n-3), promotes further growth and utilization of nutrients when Nile tilapia are maintained for long periods at low temperatures such as 22 °C [14–16]. Several studies have reported that Nile tilapia are efficient in storing body n-3 LC-PUFA when available in the diet, especially in response to cold temperature, increasing fatty acid unsaturation in cell membranes to maintain their fluidity and permeability [15,18]. Such a mechanism is reflected in an increase in body PUFA (mainly n-3 PUFA) concentration [10,18].

The main sources of n-3 LC-PUFAs in aquafeeds are fish meal and oil, but with the stagnation of extractive fishing, the use of these feedstuffs has become unsustainable for aquaculture production [19]. Currently, there is a growing number of studies looking for possible substitutes for fish meal and oil, mostly using plant meals and/or plant oils. However, the fatty acid composition of plant oils can be a limiting factor when used as an exclusive lipid source, since they are commonly deficient in n-3 PUFA, presenting a high n-6:n-3 ratio [20]. In addition, the effects of plant oils on lipid metabolism and the health of fish may lead to imbalances in body fatty acids, affecting organ integrity [21].

*Aurantiochytrium* sp. is a heterotrophic microorganism found in marine habitats, belonging to the family Thraustochytridae, which can be considered as a possible novel substitute for fish oil because it is rich in DHA. Our team evaluated the inclusion of such a novel feed additive in diets of Nile tilapia at an optimal temperature (28 °C) [22,23] and at a suboptimal temperature (22 °C) [15]. Our findings show that for Nile tilapia kept at cold, suboptimal temperature, dietary supplementation with 10 g kg$^{-1}$ *Aurantiochytrium* sp. meal (AM) improved growth by 16%, in addition to improving feed efficiency, specific growth rate, and apparent net protein retention when compared to fish fed a diet without supplementation [15]. When comparing two sources of DHA (AM and cod liver oil, CLO), at similar DHA concentrations, fish fed the diet supplemented with AM had significantly higher growth performance, feed efficiency, and protein utilization than fish fed the CLO-supplemented diet. However, the effect of supplementing this novel feed additive on the integrity and composition of important organs such as gut and hepatopancreas have not yet been evaluated in Nile tilapia. Nevertheless, histopathological changes were already reported in Nile tilapia reared at suboptimal growth temperatures [13]. Thus, in this study we aimed beyond the assessment of zootechnical gains, by evaluating body lipid profiles, somatic indices, and the histology of important organs. Such an approach will allow a

more reliable assessment of 'healthy growth' when supplementing a novel feed additive, rich in DHA, to Nile tilapia, at a suboptimal low temperature.

## 2. Materials and Methods

### 2.1. Experimental Design and Diets

The experimental diets were formulated to meet the nutritional requirements of Nile tilapia [24,25] using practical ingredients (Table 1). Five experimental diets were obtained by supplementing with AM (ALL-G-RICH™, provided by Alltech®, Nicholasville, KY, USA), at concentrations of 0.0 (un-supplemented), 5, 10, 20, and 40 g kg⁻¹ of the diet. Dietary treatments were named 0AM, 5AM, 10AM, 20AM, and 40AM, respectively. Additionally, a positive control diet was formulated, aiming at comparing two DHA-rich sources: a traditional source (CLO) versus a novel feed additive (AM). The CLO diet was only compared to the 10AM diet, since both contained the same amount of DHA in their final composition (~0.2 g kg⁻¹ dry diet).

**Table 1.** Formulation, proximate composition, and fatty acid profile of the experimental diets.

| | **Diets** | | | | | |
| --- | --- | --- | --- | --- | --- | --- |
| | **0AM** | **5AM** | **10AM** | **20AM** | **40AM** | **CLO** |
| Ingredient [a], g kg⁻¹ dry diet | | | | | | |
| Soybean meal | 473.8 | 471.6 | 472.0 | 469.9 | 465.9 | 477.4 |
| Corn | 321.7 | 320.0 | 316.0 | 315.6 | 305.0 | 314.1 |
| Poultry by-product meal | 157.2 | 157.2 | 157.2 | 157.2 | 157.2 | 157.2 |
| Vitamin and mineral premix [b] | 28.3 | 28.3 | 28.3 | 28.3 | 28.3 | 28.3 |
| Swine lard | 19.0 | 17.9 | 16.5 | 9.0 | - | - |
| Corn oil | - | - | - | - | 3.60 | 3.00 |
| ALL-G-RICH™ | - | 5.0 | 10.0 | 20.0 | 40.0 | - |
| Cod liver oil | - | - | - | - | - | 20.0 |
| Composition, g 100 g⁻¹ dry weight | | | | | | |
| Gross energy, kcal kg⁻¹ | 4168 | 4251 | 4216 | 4257 | 4297 | 4168 |
| Dry matter | 89.47 | 90.32 | 89.22 | 89.74 | 90.66 | 90.32 |
| Crude protein | 36.40 | 36.30 | 36.08 | 35.93 | 35.85 | 36.20 |
| Lipid | 8.64 | 8.93 | 8.99 | 9.13 | 9.90 | 9.20 |
| Ash | 7.11 | 7.17 | 7.18 | 7.20 | 7.61 | 7.20 |
| 16:0 PAL [c] | 1.55 | 1.63 | 1.77 | 2.03 | 2.43 | 1.16 |
| 18:1 n-9 OLA | 2.56 | 2.31 | 1.93 | 1.89 | 1.49 | 1.89 |
| 18:2 n-6 LOA | 1.92 | 1.81 | 1.75 | 1.76 | 1.71 | 1.81 |
| 20:4 n-6 ARA | ND [e] | ND | ND | ND | ND | 0.05 |
| 18:3 n-3 α-LNA | 0.04 | 0.03 | 0.03 | 0.02 | 0.02 | 0.23 |
| 20:5 n-3 EPA | ND | ND | ND | 0.01 | 0.02 | 0.17 |
| 22:5 n-3 DPA | ND | ND | ND | 0.01 | 0.02 | 0.04 |
| 22:6 n-3 DHA | ND | 0.09 | 0.20 | 0.38 | 0.75 | 0.23 |
| Σ SFA [d] | 2.14 | 2.17 | 2.24 | 2.57 | 2.98 | 1.29 |
| Σ MUFA | 3.02 | 2.72 | 2.28 | 2.26 | 1.80 | 2.65 |
| Σ PUFA | 1.98 | 2.00 | 2.04 | 2.37 | 2.70 | 2.79 |
| Σ PUFA n-6 | 1.94 | 1.86 | 1.83 | 1.89 | 1.87 | 1.89 |
| Σ LC-PUFA n-6 | 0.02 | 0.06 | 0.08 | 0.13 | 0.16 | 0.08 |
| Σ PUFA n-3 | 0.04 | 0.13 | 0.21 | 0.47 | 0.84 | 0.83 |
| Σ LC-PUFA n-3 | ND | 0.09 | 0.20 | 0.45 | 0.82 | 0.54 |
| n-3:n-6 | 0.02 | 0.07 | 0.12 | 0.25 | 0.45 | 0.44 |

[a] Corn and soybean meal provided by Nicoluzzi Rações Ltd.a (Penha, Santa Catarina, Brazil). Poultry by-product meal was produced by Kabsa S.A. (Porto Alegre, Rio Grande do Sul, Brazil).

Swine lard was produced by Seara Alimentos S.A. (Itajaí, Santa Catarina, Brazil). ALL-G-RICH™ was produced by Alltech Inc. (Nicholasville, Kentucky, USA) and imported by Alltech do Brasil Agroindustrial Ltd.a (Araucária, Paraná, Brazil). Corn oil "Suavit" was produced by Cocamar Ltd.a (Maringá, Paraná, Brazil). Cod liver oil "Möllers Tran" was produced by Orkla Health (Oslo, Østlandet, Norway). [b] Dicalcium phosphate (13.5 g kg$^{-1}$), choline bitartrate (3.0 g kg$^{-1}$), butylated hydroxytoluene (BHT) (1.0 g kg$^{-1}$), threonine (0.80 g kg$^{-1}$), and vitamin–micromineral premix (10.0 g kg$^{-1}$; produced by Cargill, Campinas, São Paulo), composition per kg: folic acid 420 mg, pantothenic acid 8333 mg, BHT 25,000 mg, biotin 134 mg, cobalt sulphate 27 mg, copper sulphate 1833 mg, iron sulphate 8000 mg, calcium iodate 92 mg, manganese sulphate 3500 mg, niacin 8.333 mg, selenite 100 mg, vitamin (vit.) A 1666,670 UI, vit. B$_1$ 2083 mg, vit. B$_{12}$ 5000 μg, vit. B$_2$ 4166 mg, vit. B$_6$ 3166 mg, ascorbic acid equivalent 66,670 mg, vit. D$_3$ 666,670 UI, vit. E 16,666 UI, vit. K$_3$ 833 mg, zinc sulfate 23,330 mg, inositol 50,000 mg, and calcium propionate 250,000 mg. [c] Fatty acids: PAL, palmitic acid; OLA, oleic acid; LOA, linoleic acid; ARA, arachidonic acid; $\alpha$-LNA, alpha-linolenic acid; EPA, eicosapentaenoic acid; DPA, docosapentaenoic acid; DHA, docosahexaenoic acid. [d] Groups of fatty acids: SFA = saturated, MUFA = monounsaturated, PUFA = polyunsaturated (>2 double bonds), LC-PUFA = long-chain PUFA (>20 carbons). [e] Not detected (<0.05% total fatty acid).

The extrusion parameters were the same as those described by Nobrega et al. [15]. After extrusion, the pellets (2 to 3 mm) were dried at 50 °C in a forced air circulation oven to 8% moisture, stored in tightly closed containers in the absence of light, and kept at 20 °C to prevent fatty acid oxidation.

The experiment was run in a completely randomized design comprised of six diets with five replicates for each dietary treatment.

### 2.2. Fish and Experimental Procedures

Nile tilapia juveniles, of GIFT (genetic improvement of farmed tilapia) lineage, sexually inverted to males, were acclimated to the laboratory conditions for five weeks in three 1000 L tanks, connected to a freshwater recirculation aquaculture system (RAS), and water temperature control (28 °C). The photoperiod was adjusted to 12 h.

Thereafter, groups of 25 fish were randomly stocked into 30 tanks with 100 L capacity and were acclimatized to experimental conditions at 28 °C for a week. In the second week, water temperature was lowered gradually from 28 °C to 22 °C (1 °C per day) and, in the third week of acclimation, water temperatures were maintained at 22 °C. During this acclimation period, fish were fed a negative control diet without supplementation of AM.

At the beginning of the trial, fish average body weight was 8.47 ± 0.19 g (average ± standard error). Fish were fed twice daily (10:00 and 18:00 h) to apparent satiation for 87 days.

### 2.3. Sample Collection

At the end of the experiment, fish were deprived of feed for 24 h and euthanized via overdose (200 mg L$^{-1}$) of the anesthetic Eugenol® (Biodinâmica Química and Farmacêutica Ltd., Ibiporã, PR, Brazil), followed by sectioning of the dorsal spine. At the beginning of the experiment, 90 fish (three groups of 30 fish) were euthanized with a lethal dose of Eugenol® and stored at −20 °C for analysis of the initial fatty acid body profile.

For fatty acid analysis and retention calculation, 15 fish per dietary treatment (three per experimental unit) were euthanized and stored at −20 °C. Additionally, the hepatopancreas and the anterior part of the intestine of 10 fish per treatment (two per experimental unit) were dissected and fixed in 10% buffered formalin for histological analyses. For the analyses of fatty acids and glycogen, the hepatopancreas of 15 fish (three fish from each experimental unit) were sampled, immediately frozen in liquid nitrogen, and stored at −80 °C until the analyses were performed.

Then, 10 fish per treatment (two fish per experimental unit) were dissected, and the viscera and hepatopancreas were weighed to calculate viscerosomatic index (VSI) and hepatosomatic index (HSI) indices, according to the following equations:

$$VSI(\%) = (\text{viscera weight} \div \text{body weight}) \times 100 \qquad (1)$$

$$HSI(\%) = (\text{hepatopancreas weight} \div \text{body weight}) \times 100 \qquad (2)$$

### 2.4. Proximate and Biochemical Composition Analyses

The proximate composition of the diets was conducted at the Fish Nutrition Lab (LABNUTRI, UFSC), following procedures standardized by the Association of Official Analytical Chemists [26]: moisture (drying at 105 °C to constant weight, method 950.01), crude protein (Kjeldahl, method 945.01), total lipid (Soxhlet, method 920.39C), and ash (incineration at 550 °C, method 942.05). Crude energy was determined in a calorimeter (PARR, model ASSY 6200), according to instructions from the manufacturer.

The fatty acid profiles of the experimental diets, fish, and hepatopancreas were analyzed using gas chromatography. The lipid extraction and chromatographic conditions were analyzed using the procedures described by Nobrega et al. [15]. Fatty acids detected and summed but not included in the tables: 10:0, 12:0, 14:0, 15:0, 18:0, 20:0, 22:0, 16:1 n-7, 18:1 n -9; 18:1 n-7, 20:1 n-9, 22:1 n-9, 24:1 n-9, 16:2 n-4, 18:3 n-6, 18:4 n-3, 20:2 n-6, 20:3 n-6, 20:4 n-3, and 22:3 n-3. The most important fatty acids for fish metabolism are included in the body and liver composition tables. The sum of the fatty acid group's SFA, monounsaturated (MUFA), PUFA, n-6 PUFA, n-3 PUFA, and LC-PUFA are also expressed. The n-3:n-6 ratio was calculated using the $\sum$n-3 PUFA:$\sum$n-6 PUFA.

Following the fatty acid profile analyses, we calculated the apparent body retention rates (ARRs) of LOA, $\alpha$-LNA, and DHA, as well as the total n-3 PUFA and n-6 PUFA groups, following the methodology proposed by Glencross et al. (2003). The equation used was:

$$ARR = 100 \times \{(FAf - FAi) \times (FAc) - 1\} \qquad (3)$$

where FAf is the absolute amount of a specific fatty acid in the fish at the end of the study, FAi is the absolute specific fatty acid in the fish at the initial time, and FAc is the total consumption of specific fatty acids over the study period.

Hepatopancreas glycogen analysis was performed at the Laboratory of Biomarkers and Aquatic and Immunochemical Contamination (LABCAI, UFSC), following the methodology proposed by Carrol et al. [27], with some modifications, including the addition of 1.0 mL of 10% trichloroacetic acid to 0.15 g of tissue, and centrifugation at 3000× *g* for 15 min at 5 °C. A 0.35 mL aliquot of the supernatant was added to previously refrigerated ethanol (1:5, v:v). The mixture was centrifuged at 5000× *g* for 30 min at 5 °C, the supernatant was discarded, and 0.2 mL of ethanol was added, and then the mixture was centrifuged again for 5 min. The pellet formed was dried at 50 °C and resuspended in 1.5 mL ultrapure water. After total dilution of the pellet, a 0.1 mL aliquot of the extract and 1.0 mL of 10% anthrone reagent were heated for 5 min at 90 °C, and the mixture was immediately immersed in ice. An aliquot of 0.2 mL of the final mixture was used for microplate reading at 620 nm in a spectrophotometer.

### 2.5. Histological Analyses

Histological analyses were performed at the Laboratory of Pathology and Health of Aquatic Organisms (AQUOS, UFSC). The organs (hepatopancreas and the anterior third of the intestines) were dehydrated in a series of increasing ethyl alcohol concentrations, clarified in xylol, and embedded in paraffin at 60 °C. Organs were cut to a thickness of 3 to 5 μm (PAT-MR10 microtome) and two cuts of each organ were evaluated. The slides were stained with hematoxylin and eosin (H&E). After staining, the slides were mounted in Entellan® medium and analyzed under a microscope, as described by Brum et al. [28].

Regarding intestinal morphology, we measured the height and width of the intestinal folds and quantified the number of folds and goblet cells using Zen Pro software (Zeiss, Jena, Germany). Histological alterations in the hepatopancreas were evaluated

semiquantitatively by ranking the severity of tissue lesions, according to the modified method described by Schwaiger et al. [29]. The ranking was 0 (absence of alteration), 1 (mild alteration, corresponding to <25% of the tissue area), 2 (moderate alteration, 25% to 50% of the tissue area), and 3 (severe alteration, >50% of the tissue area).

The following alterations were considered in the hepatopancreas: cordonal appearance and hepatocyte size variation, pancreas with intact acini, hepatocyte ballooning, cholestasis, large vessel congestion in the pancreas and sinusoids, hepatocyte nucleus displacement, sinusoid dilatation, eosinophilic and mononuclear lymphocytic, macrosteatosis, microsteatosis, necrosis, nucleus with pyknosis, karyolysis and kariorrhexis, loss of pancreatic structure, and presence of bilirubin.

### 2.6. Statistical Analyses

To determine the optimal dietary concentration of AM, the dependent variables related to body composition and retention of fatty acids, hepatic glycogen, total fat and fatty acids in the hepatopancreas, somatic index, and intestinal measurements of fold height, fold width, fold count, and goblet cell count were subjected to polynomial regression analysis. To evaluate the same variables when comparing the CLO and 10AM diets, we used the Student's t-test. The histological changes evaluated in the intestine and hepatopancreas were analyzed using the non-parametric Kruskal–Wallis test, followed by Dunn's test. For all statistical analyses, Statistica 13.0 software (Statsoft Inc. Tulsa, OK, USA) was used and a significance level of 5% was adopted.

## 3. Results

### 3.1. Body Fatty Acid Composition and Apparent Retention

The increasing dietary levels of AM significantly ($p < 0.05$) affected fatty acid body composition, presenting a significant linear response (Table 2). The body content of DHA, total PUFA, n-3 PUFA, and n-3 LC-PUFA increased linearly. Docosahexaenoic acid concentration increased from 0.02 to 0.41 g 100 g$^{-1}$ in fish fed 0AM and 40AM, respectively. The concentration of Σ LC-PUFA n-3 also ranged from 0.04 g 100 g$^{-1}$ (0AM) to 0.48 g 100 g$^{-1}$ (40AM). The body contents of eicosapentaenoic acid (EPA, 20:5 n-3) and docosapentaenoic acid (DPA, 22: 4 n-6) were detected only in fish fed with the highest concentrations of AM, 20 and 40AM. In contrast, the body concentrations of arachidonic acid (ARA, 20:6 n-6), SFA, MUFA, n-6 PUFA, and n-6 LC-PUFA decreased linearly in fish fed increasing levels of AM.

When body fatty acid composition was compared between DHA sources (CLO versus 10AM), fish fed the 10AM diet presented higher palmitic acid (PAL) content and n-6 LC-PUFA (Table 3). Conversely, fish fed the CLO diet showed higher concentrations of α-LNA, DHA, n-3 PUFA, n-3 LC-PUFA, and n-3:n-6 ratio. EPA and DPA were detected only in fish fed the CLO diet.

Adjustment of a polynomial quadratic regression was significant for the apparent retention rate (ARR) of body fatty acid ($p < 0.05$) (Table 4). According to the regression trend of DHA, there was a reduction in the body retention rate of this fatty acid with an increase in dietary levels of AM. The ARR of DHA was 57.96% and 29.59% in fish fed the lowest inclusion level (5AM) and the highest inclusion level (40AM), respectively. The percent retention of DHA decreased by approximately 28% between fish fed the highest and lowest inclusion. The diet without the inclusion of AM did not contain DHA (Table 1); therefore, the retention rate was not calculated for fish fed that dietary treatment. With increasing levels of dietary inclusion of AM, a positive tendency was observed in the ARR for α-LNA. However, ARR for n-6 and n-3 PUFA showed a negative trend, decreasing its retention with the increased dietary inclusion of AM. There was no significant difference in the ARR of LOA among fish fed the different inclusion levels of the additive.

**Table 2.** Whole body fatty acid composition of Nile tilapia juveniles fed increasing concentrations of *Aurantiochytrium* sp. meal (AM) for 87 days, at 22 °C [1].

| Fatty Acids g 100 g⁻¹ Dry Weight | Initial Fish | Diets | | | | | Pooled SEM [2] | p Value [3] |
|---|---|---|---|---|---|---|---|---|
| | | 0AM | 5AM | 10AM | 20AM | 40AM | | |
| 16:0 PAL | 0.52 | 2.09 | 2.22 | 2.02 | 2.09 | 2.02 | 0.18 | NS [4] |
| 18:2 n-6 LOA | 0.20 | 1.08 | 1.15 | 1.10 | 1.15 | 1.14 | 0.10 | NS |
| 20:4 n-6 ARA | 0.03 | 0.11 | 0.09 | 0.08 | 0.08 | 0.07 | 0.01 | <0.001 |
| 22:4 n-6 ADA | ND [5] | 0.13 | 0.10 | 0.10 | 0.10 | 0.11 | 0.02 | NS |
| 18:3 n-3 α-LNA | 0.01 | 0.05 | 0.06 | 0.06 | 0.06 | 0.06 | 0.01 | NS |
| 20:5 n-3 EPA | ND | ND | ND | ND | 0.04 | 0.04 | 0.01 | <0.001 |
| 22:5 n-3 DPA | ND | ND | ND | ND | 0.03 | 0.03 | 0.00 | <0.001 |
| 22:6 n-3 DHA | 0.02 | 0.04 | 0.09 | 0.11 | 0.23 | 0.41 | 0.02 | <0.001 |
| Σ SFA [6] | 0.78 | 3.05 | 3.21 | 2.89 | 2.78 | 2.50 | 0.26 | 0.007 |
| Σ MUFA | 0.88 | 4.31 | 4.39 | 4.08 | 4.02 | 3.67 | 0.36 | <0.001 |
| Σ PUFA | 0.27 | 1.84 | 1.87 | 1.80 | 2.02 | 2.16 | 0.16 | <0.001 |
| Σ PUFA n-6 | 0.25 | 1.83 | 1.79 | 1.63 | 1.60 | 1.55 | 0.14 | <0.001 |
| Σ LC-PUFA n-6 | 0.03 | 0.35 | 0.30 | 0.29 | 0.28 | 0.27 | 0.03 | <0.001 |
| Σ PUFA n-3 | 0.04 | 0.15 | 0.20 | 0.22 | 0.40 | 0.60 | 0.03 | <0.001 |
| Σ LC-PUFA n-3 | 0.03 | 0.04 | 0.09 | 0.11 | 0.30 | 0.48 | 0.02 | <0.001 |
| n-3:n-6 | 0.18 | 0.10 | 0.12 | 0.14 | 0.24 | 0.32 | 0.01 | <0.001 |

[1] Results are based on linear regression and expressed as the average of five replicates ($n = 3$ fish per replicate). [2] Standard error of means. [3] Where linear regression was significant, the following equations were obtained: ARA $y = -0.0091x + 0.0987$, $R^2 = 0.720$; EPA $y = 0.0119x - 0.002$, $R^2 = 0.723$; DPA $y = 0.0094x - 0.002$, $R^2 = 0.7709$; DHA $y = 0.0936x + 0.0361$, $R^2 = 0.9828$; SFA $y = -0.0566x + 3.0886$, $R^2 = 0.1346$; MUFA $y = -0.0566x + 3.0886$. $R^2 = 0.1346$; PUFA $y = 0.0866x + 1.8097$, $R^2 = 0.5196$; PUFA n-6 $y = -0.0866x + 1.8097$, $R^2 = 0.5196$; LC-PUFA n-6 $y = -0.0166x + 0.324$, $R^2 = 0.4313$; PUFA n-3 $y = 0.1157x + 0.1394$, $R^2 = 0.9656$; LC-PUFA n-3 $y = 0.1149x + 0.0321$, $R^2 = 0.9705$; n-3:n-6 $y = 0.0584x + 0.0967$, $R^2 = 0.9719$. [4] Not significant ($p > 0.05$). [5] Not detected (<0.05% of total fatty acids). [6] Groups of fatty acids: SFA = saturated, MUFA = monounsaturated, PUFA = polyunsaturated (>2 double bonds), LC-PUFA = long-chain PUFA (>20 carbons).

When comparing the two sources of fatty acids, 10AM versus CLO, the ARRs of α-LNA and n-6 PUFA were higher in fish fed 10AM (Table 5). In contrast, the ARRs of DHA and n-3 PUFA were higher in the CLO-fed fish.

**Table 3.** Whole body fatty acid composition of Nile tilapia juveniles fed two sources of docosahexaenoic acid (DHA) for 87 days, at 22 °C [1,2].

| Fatty Acids g 100 g⁻¹ Dry Weight | Diets | | p Value |
|---|---|---|---|
| | 10AM | CLO | |
| 16:0 PAL | 2.02 ± 0.14 | 1.72 ± 0.18 | 0.037 |
| 18:2 n-6 LOA | 1.10 ± 0.08 | 1.01 ± 0.07 | NS [3] |
| 20:4 n-6 ARA | 0.08 ± 0.00 | 0.08 ± 0.01 | NS |
| 18:3 n-3 α-LNA | 0.06 ± 0.01 | 0.08 ± 0.01 | <0.001 |
| 20:5 n-3 EPA | ND [4] | 0.06 ± 0.01 | - |
| 22:5 n-3 DPA | ND | 0.05 ± 0.00 | - |
| 22:6 n-3 DHA | 0.11 ± 0.01 | 0.18 ± 0.01 | <0.001 |
| Σ SFA [5] | 2.89 ± 0.21 | 2.54 ± 0.26 | NS |
| Σ MUFA | 4.08 ± 0.33 | 3.76 ± 0.34 | NS |
| Σ PUFA | 1.80 ± 0.13 | 1.83 ± 0.12 | NS |
| Σ PUFA n-6 | 1.55 ± 0.11 | 1.46 ± 0.10 | NS |
| Σ LC-PUFA n-6 | 0.29 ± 0.02 | 0.23 ± 0.02 | <0.001 |

| | | | |
|---|---|---|---|
| Σ PUFA n-3 | 0.22 ± 0.02 | 0.42 ± 0.03 | <0.001 |
| Σ LC-PUFA n-3 | 0.11 ± 0.01 | 0.29 ± 0.02 | 0.008 |
| n-3:n-6 | 0.14 ± 0.00 | 0.29 ± 0.10 | <0.001 |

[1] Results are based on a t-test and expressed as the average of five replicates (*n* = 3 fish per replicate), followed by the standard error. [2] Diets with similar contents of DHA (~0.2 g kg⁻¹ DHA dry diet); 10AM = 10 g kg⁻¹ *Aurantiochytrium* sp. meal and CLO = cod liver oil. [3] Not significant (*p* > 0.05). [4] Not detected (<0.05% of total fatty acids). [5] Groups of fatty acids: SFA = saturated, MUFA = monounsaturated, PUFA = polyunsaturated (≥2 double bonds), LC-PUFA = long-chain PUFA (≥20 carbons).

**Table 4.** Apparent body retention rate (ARR) of fatty acids in Nile tilapia juveniles fed increasing concentrations of *Aurantiochytrium* sp. meal (AM) for 87 days, at 22°C [1].

| ARR, % | Diets | | | | | Pooled SEM [2] | *p* Value [3] |
|---|---|---|---|---|---|---|---|
| | 0AM | 5AM | 10AM | 20AM | 40AM | | |
| 18:2 n-6 LOA | 38.52 | 38.29 | 37.72 | 36.02 | 36.21 | 1.83 | NS [4] |
| 18:3 n-3 α-LNA | 102.60 | 107.21 | 116.81 | 138.67 | 174.98 | 7.20 | <0.001 |
| 22:6 n-3 DHA | - | 57.96 | 34.41 | 32.75 | 29.59 | 2.09 | <0.001 |
| Σ PUFA [5] n-6 | 54.19 | 49.01 | 47.30 | 42.97 | 41.25 | 2.27 | 0.004 |
| Σ PUFA n-3 | 277.15 | 95.85 | 58.11 | 46.49 | 38.48 | 4.44 | <0.001 |

[1] Results are based on quadratic regression and expressed as the average of five replicates (*n* = 3 fish per replicate). [2] Standard error of means. [3] When the polynomial regression was significant, the following equations were obtained: α-LNA y = $0.345x^2 + 17.306x + 100.631$, $R^2$ = 0.761; DHA y = $4.909x^2 - 28.688x + 66.389$, $R^2$ = 0.687; PUFA n-6 y = $1.181x^2 - 7.802x + 232.205$, $R^2$ = 0.487; PUFA n-3 y = $33.967x^2 - 182.232x + 232.205$, $R^2$ = 0.801. [4] Not significant (*p* > 0.05). [5] Total polyunsaturated (≥2 double bonds) fatty acids.

**Table 5.** Apparent body retention rate (ARR) of fatty acids in Nile tilapia juveniles fed two sources of DHA for 87 days, at 22 °C [1,2].

| ARR, % | Diets | | *p* Value [3] |
|---|---|---|---|
| | 10AM | CLO | |
| 18:2 n-6 LOA | 37.72 ± 5.74 | 32.49 ± 1.84 | NS [4] |
| 18:3 n-3 α-LNA | 116.81 ± 19.64 | 17.47 ± 0.65 | 0.008 |
| 22:6 n-3 DHA | 34.41 ± 5.59 | 44.52 ± 2.68 | 0.007 |
| Σ PUFA [4] n-6 | 47.30 ± 6.90 | 39.32 ± 1.83 | 0.032 |
| Σ PUFA n-3 | 58.11 ± 9.18 | 287.86 ± 41.73 | <0.001 |

[1] Results are based on a t-test and expressed as the average of five replicates (*n* = 3 fish per replicate), followed by the standard error. [2] Diets with similar contents of DHA (~0.2 g kg⁻¹ DHA dry diet); 10AM = 10 g kg⁻¹ *Aurantiochytrium* sp. meal and CLO = cod liver oil. [3] Not significant (*p* > 0.05). [4] Total polyunsaturated fatty acids (≥2 double bonds).

### 3.2. Fatty Acid Composition in the Hepatopancreas

Total lipid and fatty acid composition in the hepatopancreas of Nile tilapia juveniles was influenced by dietary AM contents, presenting a significant linear response (*p* < 0.05) (Table 6). Linear regression best fitted the results. With increasing dietary AM levels, the concentration of total lipid in the liver decreased linearly, ranging from 30.69 to 23.65 g 100 g⁻¹ between the treatments 0AM and 40AM. Some fatty acids, such as ARA, ADA (adrenic acid, 22:4 n-6), EPA, n-6 PUFA, and n-6 LC-PUFA, showed a similar response to total lipids. However, DHA, n-3 PUFA, n-3 LC-PUFA, and n-3:n-6 ratio hepatopancreas lipid contents increased linearly with the inclusion of AM. The DHA content in the hepatopancreas presented a significant variation of 0.15 to 1.05 g 100 g⁻¹ between treatments 0AM and 40AM, equivalent to a 105% increase in the accumulation of DHA.

**Table 6.** Total lipid and fatty acid composition of hepatopancreas in Nile tilapia juveniles fed increasing concentrations of *Aurantiochytrium* sp. meal (AM) for 87 days, at 22 °C [1].

| Fatty Acids g 100 g[−1] Dry Weight | Diets | | | | | Pooled SEM [2] | *p* Value [3] |
|---|---|---|---|---|---|---|---|
| | 0AM | 5AM | 10AM | 20AM | 40AM | | |
| Total lipid | 30.69 | 28.88 | 27.44 | 24.43 | 23.65 | 6.30 | 0.024 |
| 16:0 PAL | 4.11 | 4.00 | 4.42 | 3.57 | 3.44 | 1.30 | NS [4] |
| 18:2 n-6 LOA | 0.72 | 1.03 | 0.99 | 0.85 | 1.02 | 0.37 | NS |
| 20:4 n-6 ARA | 0.50 | 0.51 | 0.50 | 0.46 | 0.38 | 0.09 | 0.002 |
| 22:4 n-6 ADA | 0.83 | 0.72 | 0.64 | 0.51 | 0.43 | 0.15 | <0.001 |
| 18:3 n-3 α-LNA | 0.47 | 0.67 | 0.75 | 0.47 | 0.46 | 0.24 | NS |
| 20:5 n-3 EPA | 0.09 | 0.08 | 0.08 | 0.07 | 0.06 | 0.02 | 0.008 |
| 22:6 n-3 DHA | 0.15 | 0.40 | 0.56 | 0.80 | 1.05 | 0.08 | <0.001 |
| Σ SFA [5] | 7.89 | 8.29 | 8.55 | 7.17 | 6.82 | 2.04 | NS |
| Σ MUFA | 8.21 | 8.39 | 9.36 | 7.28 | 7.06 | 2.87 | NS |
| Σ PUFA | 3.25 | 4.28 | 4.01 | 3.57 | 3.94 | 0.90 | NS |
| Σ PUFA n-6 | 2.40 | 2.93 | 2.47 | 2.12 | 2.09 | 0.66 | NS |
| Σ LC-PUFA n-6 | 1.61 | 1.91 | 1.48 | 1.27 | 1.07 | 0.33 | <0.001 |
| Σ PUFA n-3 | 0.85 | 1.35 | 1.54 | 1.45 | 1.70 | 0.32 | <0.001 |
| Σ LC-PUFA n-3 | 0.26 | 0.48 | 0.64 | 0.87 | 1.13 | 0.09 | <0.001 |
| n-3:n-6 | 0.36 | 0.46 | 0.62 | 0.69 | 0.82 | 0.11 | <0.001 |

[1] Results are based on linear regression and expressed as the average of five replicates (*n* = 3 fish per replicate). [2] Standard error of means. [3] Where linear regression was significant, the following equations were obtained: Total lipids: $y = -1.750x + 29.55$, $R^2 = 0.30$; ARA: $y = -0.0326x + 0.523$, $R^2 = 0.35$; ADA: $y = 0.774x − 0.0960$, $R^2 = 0.60$; EPA: $y = -0.00660x + 0.0858$, $R^2 = 0.30$; DHA: $y = 0.212x + 0.277$, $R^2 = 0.90$; PUFA n-3: $y = 0.162x + 1.137$, $R^2 = 0.41$; LC-PUFA n-3: $y = 0.209x + 0.360$, $R^2 = 0.91$; LC-PUFA n-6: $y = -0.174x + 1.735$, $R^2 = 0.50$; n-3:n-6: $y = 0.110x + 0.427$, $R^2 = 0.77$. [4] Not significant ($p > 0.05$). [5] Groups of fatty acids: SFA = saturated, MUFA = monounsaturated, PUFA = polyunsaturated (≥2 double bonds), LC-PUFA = long-chain PUFA (≥20 carbons).

The two different sources of DHA affected the total lipid and fatty acid profiles in the hepatopancreas (Table 7). Fish fed 10AM accumulated more lipids in the hepatopancreas. In addition, PAL, ADA, SFA, MUFA, and n-6 LC-PUFA also showed higher concentrations in fish fed 10AM. For fish fed the CLO diet, we observed higher concentrations of DHA, PUFA n-3, LC-PUFA n-3, and the n-3:n-6 ratio.

**Table 7.** Total lipid and fatty acid composition of hepatopancreas in Nile tilapia juveniles fed with two sources of DHA for 87 days, at 22 °C [1,2].

| Fatty Acids g 100 g[−1] Dry Weight | Diets | | *p* Value |
|---|---|---|---|
| | 10AM | CLO | |
| Total lipid | 27.44 ± 3.5 | 22.70 ± 1.72 | 0.026 |
| 16:0 PAL | 4.42 ± 1.09 | 3.08 ± 0.22 | 0.032 |
| 18:2 n-6 LOA | 0.99 ± 0.25 | 0.73 ± 0.05 | NS [3] |
| 20:4 n-6 ARA | 0.50 ± 0.02 | 0.48 ± 0.10 | NS |
| 22:4 n-6 ADA | 0.64 ± 0.08 | 0.32 ± 0.03 | <0.001 |
| 18:3 n-3 α-LNA | 0.75 ± 0.33 | 0.48 ± 0.09 | NS |
| 20:5 n-3 EPA | 0.08 ± 0.0 | 0.08 ± 0.01 | NS |
| 22:6 n-3 DHA | 0.56 ± 0.05 | 0.84 ± 0.04 | <0.001 |
| Σ SFA [4] | 8.55 ± 1.89 | 6.33 ± 0.38 | 0.033 |
| Σ MUFA | 9.36 ± 2.47 | 6.45 ± 0.66 | 0.034 |
| Σ PUFA | 4.01 ± 0.65 | 3.63 ± 0.29 | NS |
| Σ PUFA n-6 | 2.47 ± 0.38 | 1.65 ± 0.11 | NS |

| | | | |
|---|---|---|---|
| Σ LC-PUFA n-6 | 1.48 ± 0.07 | 1.03 ± 0.06 | 0.008 |
| Σ PUFA n-3 | 1.54 ± 0.40 | 1.90 ± 0.16 | 0.017 |
| Σ LC-PUFA n-3 | 0.64 ± 0.19 | 1.17 ± 0.14 | 0.008 |
| n-3:n-6 | 0.62 ± 0.13 | 0.87 ± 0.03 | 0.008 |

[1] Results are based on t-tests and expressed as the average of five replicates ($n$ = 3 fish per replicate), followed by the standard error. [2] Diets with similar contents of DHA (~0.2 g kg$^{-1}$ DHA dry diet); 10AM = 10 g kg$^{-1}$ *Aurantiochytrium* sp. meal and CLO = cod liver oil. [3] Not significant ($p >$ 0.05). [4] Groups of fatty acids: SFA = saturated, MUFA = monounsaturated, PUFA = polyunsaturated (≥2 double bonds), LC-PUFA = long-chain PUFA (≥20 carbons).

### 3.3. Somatic Indexes and Hepatic Glycogen Concentration

Our findings reveal that VSI, HSI, and the hepatic glycogen were not significantly affected by increasing dietary AM ($p >$ 0.05) (Table 8). Similarly, no differences were found in these same variables ($p >$ 0.05) when comparing the two sources of DHA (Table 9).

**Table 8.** Somatic indexes, hepatic glycogen, and intestinal morphometry of Nile tilapia juveniles fed increasing concentrations of *Aurantiochytrium* sp. meal (AM) for 87 days, at 22 °C [1].

| Variables | Diets | | | | | Pooled SEM [2] | *p* Value [3] |
|---|---|---|---|---|---|---|---|
| | 0AM | 5AM | 10AM | 20AM | 40AM | | |
| Viscerosomatic index | 11.76 | 11.93 | 12.10 | 12.08 | 12.26 | 0.93 | NS [4] |
| Hepatosomatic index | 2.73 | 2.95 | 2.85 | 2.76 | 2.70 | 0.48 | NS |
| Hepatic glycogen | 5.43 | 4.88 | 6.17 | 5.63 | 5.57 | 1.31 | NS |
| Intestinal morphometry | | | | | | | |
| Number of folds | 39.56 | 39.50 | 46.67 | 45.33 | 46.22 | 5.26 | 0.019 |
| Fold height, μm | 420.5 | 479.0 | 449.5 | 404.1 | 392.0 | 81.50 | NS |
| Fold width, μm | 113.2 | 119.3 | 114.3 | 110.7 | 118.6 | 22.10 | NS |
| Number of goblet cells | 362.9 | 379.6 | 428.0 | 472.0 | 324.0 | 251.90 | 0.047 |

[1] Results are based on polynomial regression analysis and expressed as the average of five replicates ($n$ = 2 fish per replicate). [2] Standard error of means. [3] When the polynomial regression was significant, the following equations were obtained: number of folds = 1.034x$^2$ + 5.806x + 39.046, R$^2$ = 0.310; number of goblet cells = −46. 423x$^2$ + 181.967x + 341.074, R$^2$ = 0.138. [4] Not significant ($p >$ 0.05).

**Table 9.** Somatic indexes, hepatic glycogen, and intestinal morphometry of Nile tilapia juveniles fed two sources of docosahexaenoic acid (DHA) for 87 days, at 22 °C [1,2].

| Variables | Diets | | *p* Value |
|---|---|---|---|
| | 10AM | CLO | |
| Viscerosomatic index | 12.10 ± 0.49 | 11.41 ± 0.71 | NS [3] |
| Hepatosomatic index | 2.85 ± 0.35 | 2.93 ± 0.20 | NS |
| Hepatic glycogen | 6.17 ± 0.66 | 5.58 ± 0.82 | NS |
| Intestinal morphometry | | | |
| Number of folds | 45.60 ± 1.60 | 41.60 ± 1.60 | NS |
| Fold height, μm | 449.56 ± 20.29 | 440.81 ± 17.58 | NS |
| Fold width, μm | 114.38 ± 4.30 | 120.04 ± 5.86 | NS |
| Number of goblet cells | 428.00 ± 97.43 | 400.36 ± 273.50 | NS |

[1] Results based on t-tests and are expressed as the average of five replicates ($n$ = 2 fish per replicate), followed by the standard error. [2] Diets with similar contents of DHA (~0.2 g kg$^{-1}$ DHA dry diet); 10AM = 10 g kg$^{-1}$ *Aurantiochytrium* sp. meal and CLO = cod liver oil. [3] Not significant ($p >$ 0.05).

### 3.4. Morphology and Histological Changes in the Intestine and Hepatopancreas

The intestinal morphology data of fish fed increasing levels of AM were evaluated using polynomial regression. A quadratic trend was observed regarding the number of intestinal folds and the supplementation with AM, which increased from 39.5 intestinal folds in fish fed 0AM and 5AM to 46.6 in fish fed 10AM (Table 8, Figure 1B,F). The inclusion of AM did not influence the height and width of the intestinal folds. However, the number of goblet cells also increased with the addition of levels up to 20AM; however, in fish fed 40AM there was a decrease in goblet cells (Table 8, Figure 1G, I). No differences were found in the morphometric variables when comparing the two sources of DHA (Table 9).

In the hepatopancreas, the following variables were not affected by increasing supplementation of AM or by feeding the CLO diet, presenting only mild to moderate changes in the tissue: cordonal aspect, intact acini, balloon aspect of hepatocytes, cholestasis, congestion in large vessels, congestion in the pancreas and sinusoids, displacement of the nucleus of hepatocytes, dilatation of the venous sinus, mononuclear and eosinophilic infiltrates, hypertrophy of hepatocytes, and nucleus with pyknosis. However, fish fed the 0AM diet presented the highest hepatocyte size variation, whereas those fed 40AM presented the lowest (Table 10, and Figure 2A,C,D). The intensity of macrosteatosis was highest in fish fed 0AM and did not differ significantly between fish fed 5AM and 10AM. Fish fed 20AM, 40AM, and CLO diets showed a low intensity of macrosteatosis, and they differed significantly from 0AM-fed fish. Fish fed the 0AM diet were the only ones showing microsteatosis in the hepatopancreas (Figure 2A). Hepatopancreas necrosis was significantly lower in fish fed 40AM than in those fed the diet 0AM (Figure 2B).

Fish fed the 5AM diet showed the lowest number of hepatocyte nuclei with karyolysis and differed from fish fed both the 10AM and 20AM diets (Figure 3A and 3C, respectively), which presented the greatest intensity of this alteration. Meanwhile, fish fed the 5AM diet also had less intensity of karyorrhex than fish fed both the 20AM and 40AM diets (Figure 3B). The loss of the nuclei by hepatocytes was more intense in fish fed the 0AM diet. Loss of the nuclei of the pancreatic acinos was observed in fish fed the diets 5AM and 10AM (Figure 3D,E). Only fish fed the 5AM and 10AM diets showed loss of the pancreatic acini nuclei (Figure 3F). The presence of macrophages with bilirubin was identified only in fish fed the CLO diet (Figure 3F).

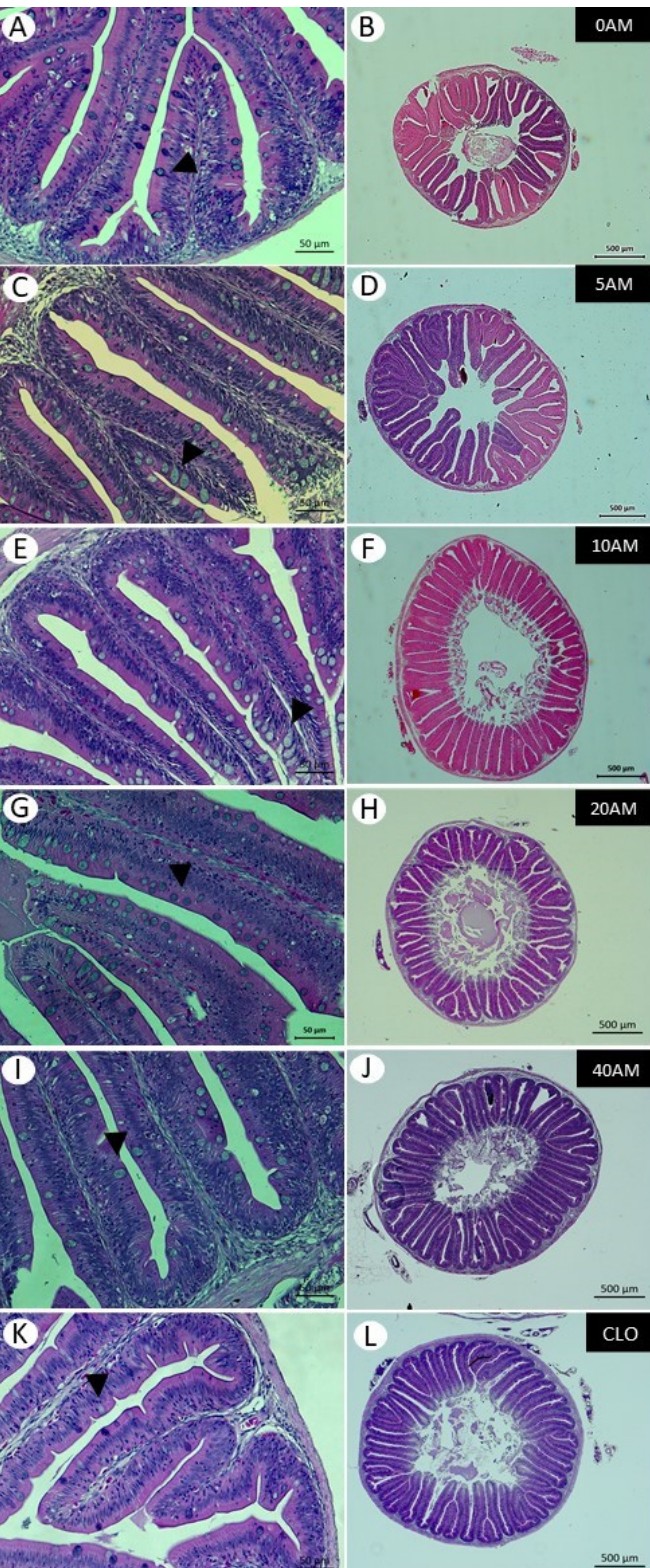

**Figure 1.** Histological changes in the anterior third of the intestines of juvenile Nile tilapia when fed different levels of AM or CLO, for 87 days, at 22 °C. Figures (**A,C,E,G,I,K**) represent a partial cross-section of the intestine of fish fed 0AM, 5AM, 10AM, 20AM, 40AM, and CLO diets, respectively. Goblet cells are identified with the arrowhead. Bar: 50 μm. Figures (**B,D,F,H,J,L**) represent a cross-section of the whole intestine. Bar: 500 μm. Staining: H&E.

**Table 10.** Morphological alterations in hepatopancreas of Nile tilapia juveniles, when fed with increasing concentrations of *Aurantiochytrium* sp. meal (AM) or cod liver oil (CLO) for 87 days, at 22°C [1].

| Variable | Diets | | | | | | *p* Value [2] |
|---|---|---|---|---|---|---|---|
| | 0AM | 5AM | 10AM | 20AM | 40AM | CLO | |
| Cell size variation | 1.83 ± 0.69 [a] | 1.00 ± 0.49 [ab] | 0.92 ± 0.41 [ab] | 1.08 ± 0.76 [ab] | 0.45 ± 0.66 [b] | 1.10 ± 0.30 [ab] | 0.0001 |
| Hypotrophy of hepatocyte nucleus | 1.00 ± 0.0 [a] | 0.08 ± 0.28 [b] | 0.00 ± 0.0 [b] | 0.00 ± 0.0 [b] | 0.00 ± 0.0 [b] | 0.00 ± 0.0 [b] | <0.0001 |
| Macrosteatosis | 1.50 ±1.16 [a] | 0.92 ± 0.9 [ab] | 1.08 ± 0.7 [ab] | 0.42 ± 0.67 [b] | 0.36 ± 0.82 [b] | 0.54 ± 0.82 [b] | 0.0288 |
| Microsteatosis | 0.83 ± 0.85 [a] | 0.00 ± 0.0 [b] | 0.00 ± 0.0 [b] | 0.00 ± 0.0 [b] | 0.00 ± 0.0 [b] | 0.00 ± 0.0 [b] | <0.0001 |
| Necrosis | 1.92 ± 0.28 [a] | 1.75 ± 0.74 [ab] | 1.58 ± 0.67 [ab] | 1.33 ± 0.65 [ab] | 1.18 ± 0.60 [b] | 1.45 ± 0.68 [ab] | 0.0376 |
| Nuclei with karyolysis | 1.58 ± 0.51 [ab] | 1.17 ± 0.39 [b] | 1.92 ± 0.67 [a] | 1.92 ± 0.29 [a] | 1.81 ± 0.60 [ab] | 1.81 ± 0.60 [ab] | 0.0066 |
| Nuclei with karyorrhexis | 1.58 ± 0.51 [bc] | 1.41 ± 0.51 [c] | 1.83 ± 0.58 [abc] | 1.92 ± 0.29 [ab] | 2.09 ± 0.54 [a] | 1.82 ± 0.60 [abc] | 0.0419 |
| Loss of hepatocyte nucleus | 0.92 ± 0.29 [a] | 0.25 ± 0.45 [b] | 0.17 ± 0.58 [b] | 0.25 ± 0.62 [b] | 0.00 ± 0.00 [b] | 0.27 ± 0.65 [b] | <0.0001 |
| Loss of nucleus in pancreatic acini | 0.00 ± 0.00 [b] | 0.58 ± 0.67 [a] | 0.25 ± 0.45 [a] | 0.00 ± 0.0 [b] | 0.00 ± 0.00 [b] | 0.00 ± 0.00 [b] | 0.015 |
| Macrophage with bilirubin | 0.00 ± 0.00 [b] | 0.00 ± 0.0 [b] | 0.00 ± 0.00 [b] | 0.00 ± 0.00 [b] | 0.00 ± 0.00 [b] | 0.55 ± 0.52 [a] | <0.0001 |

[1] Histological alterations in the hepatopancreas were evaluated semi-quantitatively by ranking the severity of the tissue lesions. The ranking was 0 (absence of alteration), 1 (mild alteration, corresponding to <25% tissue area), 2 (moderate alteration, 25% to 50% tissue area), and 3 (severe alteration, >50% tissue area). [2] Results are based on the non-parametric Kruskal–Wallis test and expressed as the average of five replicates (*n* = 2 fish per replicate), followed by the standard error. [a,b,c] Values followed by different letters are significantly different (*p* < 0.05).

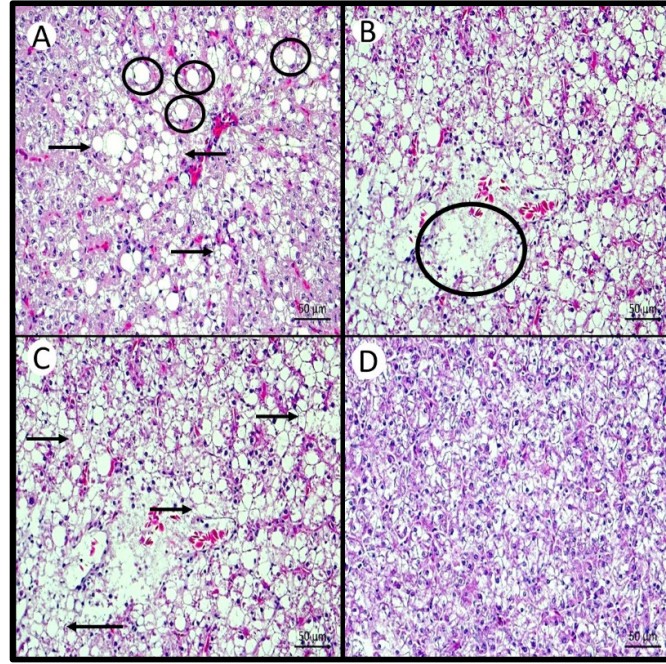

**Figure 2.** Histological changes in the hepatopancreas of Nile tilapia juveniles fed different levels of *Aurantiochytriu*m sp. meal (AM) or a cod liver oil (CLO)-supplemented diet for 87 days, at 22 °C. (**A**,**B**) High intensity of macrosteatosis (circle) and microsteatosis (arrow) in fish fed the 0AM diet. (**B**) High intensity of necrosis (circle) in fish fed the 0AM diet. (**C**) High loss of hepatocyte nucleus (arrow) in fish fed the 0AM diet. (**D**) Smaller variation in hepatocyte size in the hepatopancreas and lower intensity of macrosteatosis, microsteatosis, and necrosis in fish fed the 40AM diet.

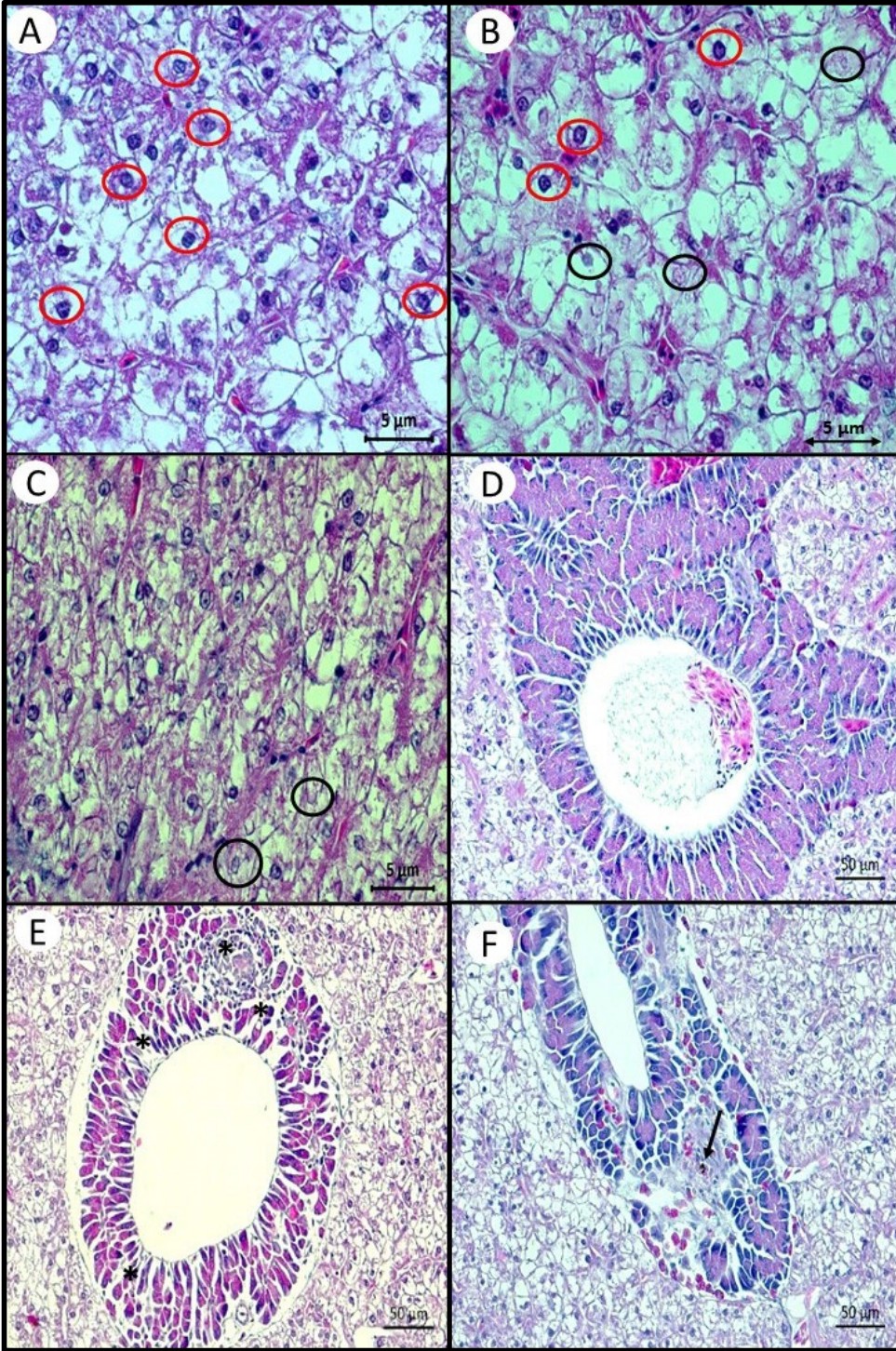

**Figure 3.** Histological changes in the hepatopancreas of Nile tilapia juveniles fed different levels of *Aurantiochytriu*m sp. meal (AM) or a cod liver oil (CLO)-supplemented diet for 87 days, at 22 °C. (**A**) High intensity of nuclei with karyolysis (red circle) in fish fed the 20AM diet. (**B**) High intensity of nuclei with karyorrhexis (black circle) in fish fed the 40AM diet. Nuclei with karyolysis are indicated by red circles. (**C**) Lower intensity of nuclei with karyolysis and karyorrex in fish fed the 5AM diet. (**D**) Nuclei in pancreatic acini without alterations in fish fed the 0AM diet. (**E**) Loss of nuclei in pancreatic acini in fish fed the 5AM diet (alteration indicated by asterisks). (**F**) Macrophage with bilirubin in fish fed the CLO diet (alteration indicated by arrow).

## 4. Discussion

### 4.1. Body Fatty Acid Composition and Apparent Retention

It has been well established in fish that the tissue fatty acid composition generally reflects dietary fatty acid composition [30–32]. Based on the regression analysis, the higher the inclusion of AM in the diets, the higher the accumulation of DHA in the whole body of the fish. Our findings also show an increase in the whole-body n-3 fatty acid content, followed by an increase in the dietary inclusion of AM, a source of DHA. Several authors cited an accumulation of DHA in the body and muscle composition of fish, especially at a low temperature [15,18,33,34]. The DHA is considered a fatty acid of high biological value, with a close link between DHA and the composition of structural phospholipids in cells [17]. Thus, DHA has a different catabolism from other fatty acids, being more preserved in cell phospholipids and less catabolized to produce energy [35]. Due to its high flexibility provided by its chemical arrangement, DHA maintains a cell membrane bilayer with the balance between fluidity and rigidity needed to accommodate rapid conformational changes in cell membrane proteins. This structure brings fluidity to the membrane at low temperatures [17]. In addition to preserving DHA, which can be modulated directly by the diet, fish also have desaturase and elongate enzymes that are responsible for desaturating LOA to ARA, LNA to EPA, and EPA to DHA. Indeed, Nile tilapia have been reported to express the genes for Δ-4 and Δ-6 desaturases, thus enabling the conversion of EPA into DHA by two different pathways, which are mediated by both desaturases [17,36].

On the other hand, during the β-oxidation of fatty acids, fish preferentially use SFA and MUFA, with shorter chain lengths and numbers of unsaturation, as energy sources [37,38]. This observation may explain the linear reduction in PAL, SFA, and MUFA in fish fed 10AM, 20AM, and 40AM, which in turn, could be preferentially degraded for energy production in order to preserve the fatty acids of greatest biological value, such as PUFAs. The decrease in ARA, n-6 PUFA, and n-6 LC-PUFA groups can be explained by the competition of desaturase and elongase enzymes, which compete for the substrates of n-3 and n-6 series fatty acids and have a greater affinity for n-3 PUFA substrates [39]. Thus, the reduction in bioconversion from n-6 PUFA to n-6 LC-PUFA may have occurred because of the inhibition of the enzymatic activity of desaturases and elongases in the presence of higher levels of DHA in the diets [32,39].

In the present study, the higher concentration of DHA in the bodies of fish fed the CLO diet than in fish fed the diet containing 10AM probably occurred due to the bioconversion of its precursors (LNA, EPA, and DPA) which were detected at higher levels only in fish fed the CLO diet. Through the study of the fatty acid balance, it was possible to understand that for Nile tilapia, the bioconversion of LNA to DHA is efficient, and the biosynthesis of DHA from EPA can be more direct or faster than the production of EPA from LNA [32]. Our study corroborates such findings, where the CLO diet contained more LNA and EPA in its composition, favoring their bioconversion to DHA.

The ARR of DHA and n-3 PUFA was influenced by the levels of dietary inclusion; the higher the dietary concentration of these fatty acids, the lower the retention. Other authors have also reported a similar pattern, where the higher the concentration of a fatty acid in the diet, the lower its relative deposition [18,39]. In our study, as also reported by Brignol et al. [22] at 28 °C, the ARR of α-LNA increased with the inclusion of AM. Increasing the dietary concentration of DHA following the inclusion of AM possibly prevented the elongation and desaturation of α-LNA for the production of LC-PUFA, thus preserving and retaining α-LNA in the membranes. In addition, diets containing higher inclusions of AM contained less α-LNA content. Consequently, the lower the presence of an essential fatty acid in the diet, the higher its body retention. However, contrary to the report by Brignol et al. [22], at 28 °C, where there was an increase in n-6 PUFA body retention with the inclusion of AM, we did not observe such a response in our study, at 22 °C. Here, PUFA n-6 retention decreased significantly with the inclusion of AM,

following the body composition trend that also decreased, probably due to the competition of desaturases and elongases with the n-3 substrate and/or due to β-oxidation of the PUFA n-6, in addition to the greater conservation of the n-3 fatty acids to maintain membrane functionality at low temperatures

When comparing the two sources of DHA, fish fed CLO showed higher retention of the following fatty acids: LOA, α-LNA, DHA, n-3 PUFA, and n-6 PUFA at 28 °C [22]. However, in our study, at 22 °C, the retention of α-LNA and n-6 PUFA was higher in fish fed the AM-supplemented diet. The higher retention of α-LNA can be explained by the low content of this fatty acid in the diet supplemented with AM, which may also have led to an increased retention of n-6 PUFA, as there was little n-3 substrate for inhibition of the desaturases and elongases in the n-6 pathway. Despite having similar amounts of DHA, the higher DHA, total PUFA, and n-3 PUFA retention in fish fed the CLO diet was probably due to the higher synthesis of DHA from precursor fatty acids, detected only in the CLO diet [17,22].

### 4.2. Fatty Acid Composition in the Hepatopancreas

The greater the dietary inclusion of AM, the lower the amount of total lipid in the liver, in addition to EPA, ARA, ADA, PUFA n-6, and LC-PUFA n-6. However, the increasing inclusions of AM increased the composition of DHA, PUFA, LC-PUFA n-3, and the n-3:n-6 ratio. The fatty acid composition of hepatopancreas largely reflected that of the diet, which is consistent with studies in Atlantic salmon [40].

The hepatopancreas is the main organ that regulates lipid metabolism, including both the synthesis and degradation of fatty acids, where several regulating enzymes show varied affinities for the different fatty acids available in that organ [37,41]. In addition, the hepatopancreas functions as an important energy reservoir, often in the form of triacylglycerols [21]. The high intake of PUFAs (mainly EPA and DHA) prevents the accumulation of lipids by inducing lipid oxidation, [41,42], inhibiting lipogenic metabolism, and stimulating the synthesis of lipoproteins [43–46].

The significant increase in DHA in the hepatopancreas, about 105% between 0AM- and 40AM-fed fish, suggests the important role of this fatty acid in cell membrane function at a low temperature. Additionally, the decrease in EPA content in the hepatopancreas may be linked to the bioconversion of EPA to DHA [17,32]. In addition, the content of n-6 PUFA decreased, suggesting the preferential route of activity of the desaturase and elongase enzymes by the n-3 series fatty acids. Indeed, Chen et al. [32] reported that increasing dietary inclusion of LNA in Nile tilapia raised at the optimal temperature could block or at least slow down n-6 LC-PUFA biosynthesis from LOA. Therefore, there is competition for accessing the Δ-6 desaturase and elongase between substrates of the n-3 and n-6 series. Although we have not evaluated such enzymes in our study, this substrate competition is well known for tilapia [32].

### 4.3. Morphology and Histological Changes in the Intestine and Hepatopancreas

Different levels of dietary supplementation with AM affected the histology of the hepatopancreas and intestine. Fish fed the highest levels of AM showed a significant increase in the number of intestinal folds. The increase in the absorption area, caused by the increase in the number of folds, could improve the digestive and absorptive processes, suggesting a more efficient use of nutrients [47]. Our results regarding the increase in the number of villi in fish fed the 10AM diet corroborate the findings of Nobrega et al. [15], where fish fed the 10AM diet reached the highest weight gain. Such high growth may be associated with an increase in the nutrient absorption area. The influence of diet-derived substances on intestinal epithelial function, including barrier integrity, is likely to be important [48].

A recent review on mammals [48] reported that n-3 LC-PUFA, especially DHA, contributes to maintaining the integrity of the intestinal epithelial barrier by exerting anti-inflammatory effects and accelerating recovery from intestinal inflammation. However,

few studies have addressed the relationship between dietary fatty acids, especially LC-PUFA n-3, and intestinal health in fish. Dietary supplementation with *Schizochytrium* sp. (a DHA-producing marine heterotrophic microorganism from the same family as *Aurantiochytrium* sp.) in juvenile mirror carp (*Cyprinus carpio* var. specularis) resulted in a higher intestinal fold height in fish fed 30 and 60 g kg$^{-1}$ supplementation if compared to fish fed the non-supplemented diet [49]. The dietary replacement of fish oil and fish meal by heterotrophic microorganisms results in a good response to intestinal integrity, since such microorganisms also contain additional bioactive cell wall compounds such as β-glucans, β-carotenes, flavonoids, nucleotides, and water-soluble peptides [50–52] which can affect nutrient availability and growth performance, but also enhance the well-being of fish by improving gut health and thus nutrient assimilation and immune competence [52].

The rising levels of AM also resulted in a significant increase in the number of goblet cells in fish fed diets 0AM to 20AM. An increase in the number of goblet cells was also reported in the intestine of Atlantic salmon, reared at an optimal temperature for the species (10.2 °C), with an increase in dietary inclusion of 6 to 15 g kg$^{-1}$ *Schizochytrium* sp. meal [52]. Goblet cells produce mucus, which plays an important role in immunity. Besides serving as a mechanical barrier, making it difficult for pathogenic bacteria to adhere, mucus contains several components of the innate immune response, such as lysozymes, immunoglobulins, complement system proteins, lectins, and several other antimicrobial components [53]. However, in fish fed the highest AM level (40AM), there was a reduction in the number of goblet cells. A previous study evaluated different dietary sources of n-3 LC-PUFA (fish oil, EPA-enriched oil, and DHA-enriched oil) for the carnivorous marine fish pompano, *Trachinotus ovatus*, raised at an optimal temperature and reported that high dietary EPA or DHA levels caused a depressed expression of Muc13 mRNA in the intestines [54]. Mucins are the major constituent of the mucous layer and are produced by goblet cells [55]. Therefore, more research needs to be carried out to assess how dietary DHA interacts with mucus production in the intestine. Furthermore, the decrease in goblet cells does not seem to be related to inflammation in the intestine, since treatments with higher AM inclusions showed a greater number of villi, showing its positive effect on intestinal morphometry.

Histological analysis of the hepatopancreas showed a similar pattern to that observed in the total lipid content of this organ. Likewise, the intensity of macrosteatosis, microsteatosis, necrosis, and loss of the nucleus of hepatocytes were lower in the hepatopancreas of fish fed the highest inclusions of AM. Therefore, the highest intensity of histological and lipid composition changes in fish fed the control diet, without the inclusion of AM but with a lipid base of corn oil and swine lard, corroborates the findings for other fish species when fish oil is replaced with vegetable oils [21,56]. Lima de Andrade et al. [13] also found histopathological changes in tilapia juveniles fed diets with different LOA:LNA (n6/n3 = 12.02 and n6/n3 = 3.85), at temperatures of 30 and 20 °C. The frequency of cell displacement injury was lower in fish fed the diet containing the low n6/n3. This may be associated with lower lipid deposition. In addition, the liver of fish fed n6/n3 = 3.85 showed low cytoplasmic vacuolization at both temperatures [13]. Therefore, an imbalance in dietary fatty acids can modify the function and morphology of this organ. It is possible to establish a relationship between the type of dietary fatty acids and the appearance of steatosis, that is, LOA > LNA > oleic acid [21].

In a three-month trial, the marine fish, *Sparus aurata*, fed diets based on fish oil, canola, flaxseed, or a mix of these oils, showed uniformity in cell size and little lipid accumulation. However, fish fed soybean oil presented steatosis foci, with hepatocytes containing numerous lipid vacuoles [21]. In another study testing rearing temperatures (optimum 12 °C and suboptimal 5 °C) and lipid sources (soy oil replacing fish oil, at 50% and 100%) in Atlantic salmon, the suboptimal temperature positively influenced the deposition of fat in liver cells and the intestines [56]. In addition, the diet with 100% soy oil resulted in the highest accumulation of fat in the liver at 5 °C [56]. According to the authors, the accumulation of fat

at low temperatures could be explained by the reduction in the activity of the enzymes involved in the esterification of fatty acids into triacylglycerol and phospholipids for very low-density lipoprotein (VLDL) production.

On the other hand, despite appearing at a low intensity, the presence of nuclei with karyolysis and karyorrhexis was lower in fish fed the lowest level of AM (5AM). It is possible that a minimal inclusion of n-3 PUFA would be sufficient to prevent these changes. Karyolysis is the complete dissolution of chromatin in a cell that is dying due to enzymatic degradation, and is usually associated with karyorrhexis, which occurs mainly as a result of necrosis [57]. This is the first time that such a change has been reported with an increase in dietary PUFAs. These changes have already been described, in greater intensity than found in our study, for several species of fish, when exposed to contaminating agents [58–60].

Several studies have shown that a drop in temperature causes fasting induced by cold, heat stress, and metabolic depression in fish [61]. To mitigate these effects, adequate nutrition is suggested in this study. We verified that AM, as a source of DHA in Nile tilapia, provides the body with the accumulation of DHA, an important fatty acid for adequate metabolic functioning of fish, when subjected to low suboptimal temperatures. In addition, increasing the inclusion of AM decreased the hepatopancreas lipid content, increased the hepatopancreas concentration of n-3 fatty acid series, and promoted significant improvements in the morphophysiology of the hepatopancreas, preventing signs of macrosteatosis, microsteatosis, and necrosis, seen in fish fed a practical diet, without any DHA supplementation. These data provide evidence for the physiological need for DHA supplementation in Nile tilapia diets at suboptimal temperatures and the potential for the development of specific winter aquafeeds for the species, with the aim of improving growth performance and physiological well-being.

In addition to its important role in fish performance and metabolism, DHA appears to play an important role in the cardiac, cardiovascular, brain, and visual functions of humans [62]. Currently, a greater intake of n-3 fatty acids is desirable for reducing the risk of many of the highly prevalent chronic diseases in Western societies, as well as in developing countries. *Aurantiochytrium* sp. meal supplementation in Nile tilapia diets can also provide a source of n-3 fatty acids for human consumption, adding value to this freshwater species.

More research should be carried out to further evaluate the effects of dietary AM supplementation in the intestinal functions of fish. We have registered a positive effect on intestinal health; however, a wide range of analyses including intestinal microbiota composition and gut-associated lymphoid tissues (GALT) immune responses would provide further understanding about the effect of such a novel additive. Furthermore, future studies should evaluate the dietary supplementation of AM in growing-out conditions to validate our findings in a controlled lab situation.

## 5. Conclusions

Nile tilapia maintained at 22 °C responded linearly to increasing dietary inclusions of AM regarding body composition. The inclusion level of 40 g kg$^{-1}$ of AM promoted the best results in improving the body and hepatopancreas n-3 fatty acid profile, decreasing hepatopancreas lipid content, and significantly improving the morphophysiology of the hepatopancreas. When comparing the two DHA sources, CLO allowed the highest n-3 fatty acid body content and retention, possibly due to the increased synthesis of DHA from fatty acid precursors. By representing a novel and renewable DHA source, the positive biological responses of Nile tilapia to dietary supplementation with AM make this additive an excellent candidate to replace fish oil in winter diets.

**Author Contributions:** Conceptualization: R.O.B., R.O.N., D.D.S., and D.M.F.; methodology: R.O.B., R.O.N., D.D.S., and D.M.F.; software: R.O.B.; validation, R.O.B. and R.O.N.; formal analysis: R.O.B.; investigation, R.O.B. and R.O.N.; resources: D.M.F. and J.E.P.; data curation: R.O.B., R.O.N., D.D.S., J.E.P., and D.M.F.; writing—original draft preparation: R.O.B.; writing—review and editing: R.O.N., D.D.S., J.E.P., and D.M.F.; supervision: D.D.S. and D.M.F.; project administration, D.M.F.; funding acquisition, D.M.F. All authors have read and agreed to the published version of the manuscript.

**Funding:** Alltech Inc. (USA), Coordenação de Aperfeicoamento de Pessoal de Nível Superior (CAPES-Brazil, Finance Code 001), and National Council for Scientific and Technological Development (CNPq-Brazil, grant # 313185/2020-4)

**Institutional Review Board Statement:** The study was conducted according to the guidelines of the National Council for Control of Animal Experimentation (CONCEA) and was approved by the Ethics Committee on Animal Use of the Federal University of Santa Catarina (CEUA/UFSC), protocol code 5665210917 in the meeting of 03/14/2018.

**Informed Consent Statement:** Not applicable.

**Data Availability Statement:** The data that support the findings of this study are available from the corresponding author upon reasonable request.

**Acknowledgments:** We thank Alltech Inc. for funding this study (USA). The additive evaluated was produced and sold by Alltech for use in animal diets under the tradename ALL-G-RICH™. We also thank CAPES-Brazil (Finance Code 001) for granting fellowships to the first and second authors and CNPq-Brazil, which granted a fellowship to the last author. Thanks are also due to Nicoluzzi Rações Ltd.a (Penha, Santa Catarina, Brazil) and Kabsa Exportadora S.A. (Porto Alegre, RS, Brazil) for donating ingredients for the manufacturing of the experimental diets, as well as to Lucas Cardoso NEPAQ-UFSC-Brazil for the assistance with the histological analysis and Jacó Joaquim Mattos LABCAI-UFSC-BRAZIL for the assistance with the glycogen analysis.

**Conflicts of Interest:** This study was funded by Alltech, Inc. (USA) through a joint research alliance between Alltech (USA) and the Universidade Federal de Santa Catarina (UFSC-Brazil). One of the authors is a consultor for Alltech through a committee containing both Alltech staff and UFSC staff, where the research project was developed and approved.

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
