# Peer review of "Aurantiochytrium sp. Meal Improved Body Fatty Acid Profile and Morphophysiology in Nile Tilapia Reared at Low Temperature"

_fishes, doi:10.3390/fishes6040045_

Round 1

Reviewer 1 Report

The authors improved the quality of the MS, but the statistical analysis should be updated.

  • The authors should study first what the polynomial regression analysis means, and then they can see that both regression and ANOVA work well!!
    Please see this paper and see how you can nicely report both ANOVA and regression.

https://doi.org/10.1016/j.aquaculture.2019.734281

Also, no need to report the R² and equations in the footnote of tables. Please simply report the P-value of each like the above paper and then discuss the results. When there is linear relation, you can say more levels should be tested to get the peak/plateau. If the P-value quadratic is significant, it means you can even introduce max or min numbers. Sometimes we need to know which treatment had the highest numbers and which had significant differences, and for these cases, ANOVA will work and not regression.

Please update the results and Tables, and I will review the next version.
Best regards

Author Response

Dear Reviewer,

We thank you for the thorough correction of our manuscript and sound contributions. However, we cannot agree with you in that aspect. Using ANOVA to analyze our data is inappropriate since multiple levels of an independent variable were tested. Therefore, we ask you to reconsider and allow us to present our results using regression analyses.

We have based our rebuttal on the references below:

“This approach (using ANOVA) ignores that response variables are continuous rather than discrete. Point-by-point analyses such as ANOVA harvest much less than the total amount of information from the data. Regression models are more effective at gleaning information from data.”

Pesti, G.M. et al. 2009. A comparison of methods to estimate nutritional requirements from experimental data. British Poultry Science, 50(1):16-52. https://doi.org/10.1080/00071660802530639

“When ANOVA is used, nutrient levels are treated as discrete rather than continuous, so that the optimum nutrient level is stated as a range between two input levels.” This is not the case in our study, where graded levels of AM were tested.

Shearer, K.D. 2000. Experimental design, statistical analysis and modelling of dietary nutrient requirement studies for fish: a critical review. Aquaculture Nutrition, 6: 91–102. https://doi.org/10.1046/j.1365-2095.2000.00134.x.

Chapter 8, pages 237-284:

Bhujel, R.C. 2008. Statistics for Aquaculture. Ames, Willey-Blackwell, 376 p.

Reviewer 2 Report

The authors responded to most of my comments. I still have the following minor concerns:

1) Line 431: Replace ” arrows” with “arrowheads”

2) Fig.1: Images G and I showed higher magnification than the A, C, E, K, however you put scale bars with the same length. In the legend, you repeated Cross-section and added the name of the group which was already inserted in the images so I suggest delete the repeated words and again add the name of stain and the value of scale bars in the legend. I know the scale bars values were provided in the images but they are minutes.

3) In the first round of revision, I asked you “

Provide supported images for what you stated in lines 210-214. Higher magnification images are necessary to confirm that.”

And you answered as follows “Indeed, the referred slides were evaluated in higher magnification. However, since there were no significant changes among fish fed the different dietary treatments, we opted for not showing such images.”

My comment: This is not acceptable in scientific research any data even those non-significant could be presented either in the main text or as supplementary. So You have two choices either delete them from text or provided the supported images as, at least, supplementary.

4) Lines 630-641: You said “high dietary EPA or DHA levels caused a depressed expression of Muc13 mRNA in the intestines [51]. The  authors suggest that dietary enrichment with EPA and DHA facilitated the intestinal  mucosal immunity barrier of fish.” It is well known that higher production of mucin is correlated with increased gut immunity. However, here you try to say that lower mucin is associated with higher gut immunity. I do not agree with this hypothesis and you should defend it with other articles from the literature. Also,  protein expression sometimes differs from its gene.

5) In my previous comment “Is it anterior third or proximal third? Previously you said anterior which is the most appropriate term.” You said “In the proximal third. We have corrected…” Based on anatomical terms of Tilapia, the intestine runs horizontally, rather than vertically, so it is incorrect to say proximal third and it could be anterior or cranial third.

Reviewer 3 Report

Manuscript ID: fishes-1398253

Title: “Aurantiochytrium sp. meal improved body fatty acid profile and morphophysiology in Nile tilapia reared at low temperature.” for Fishes.

1) General comments:

This MS describes effects of dietary supplementation of Aurantiochytrium sp. meal, rich in DHA, somatic indices, and histology of important organs of Nile tilapia reared at suboptimal low temperature (22°C) to know an adequate winter diet for Nile tilapia aquaculture. This study is interesting to hatchery managers and aquaculture researchers. Results of this study provide a reliable assessment of 'healthy growth' when supplementing a feed additive, rich in DHA, to Nile tilapia at suboptimal low temperature, which would contribute to development of optimal rearing method for this species.

Authors improved their manuscript nicely, according to reviewer comments. 

Author Response

Dear Reviewer

Thank you very much for your contribution to improve the quality of our manuscript.

Best regards,

Débora M. Fracalossi

Round 2

Reviewer 1 Report

The authors improved the quality of the MS and I suggest authors reading one more time to fix few language errors. Then, it would be ready for the final steps of acceptance.

This manuscript is a resubmission of an earlier submission. The following is a list of the peer review reports and author responses from that submission.

Round 1

Reviewer 1 Report

The authors investigated the effect of dietary supplementation of Aurantiochytrium sp. meal on body fatty acid profile, gut and hepatopancreas, morphophysiology of Nile tilapia reared at suboptimal low temperature. They designed five treatments to test the optimum dosage. This manuscript (MS) was clearly written and easy to understand. This work can help the sustainability of this species farming as few studies on this topic have been done. However, some major issues significantly compromised the quality of this MS.

Major comments:

  • The statistical analysis should be updated with polynomial regression. It is possible that some factors have quadratic relations which the results, discussion, and conclusion can be totally different. Please update the MS with the change.
  • Tables 2, 4, 6, and 8, I did not understand why you average the five treatments and compare them with Control. What was the purpose?. Some important parameters such as Σ LC-PUFA n-3 is hugely different between 5AM (0.09) and 40AM (0.48). Averaging such results can cause misleading. In Tables 1, 3, 5 and 7, please add a column for fish oil. Although you want to do the regression with five treatments, it is no harm to have fish oil there as well.
  • I will review the MS after the changes.

However, I have touched on some more points that can contribute to the improvement of this MS.

Minor comments

Abstract

  • Line 14-16, please delete it; otherwise, you need Ref for that which is not correct for the abstract.
  • It is better to start the abstract with a sentence about why you analyzed this work or something about the importance of Aurantiochytrium meal.
  • Line 18-19 is confusing. What is the positive Control?.you can simply say the Control throughout the MS. Please clearly mention what all treatments and their name were here.
  • I suggest naming treatments like 0AM, 5AM, 10AM, 20AM, and 40AM in the text, Tables and Figures.
  • Please reorder the keywords alphabetically and capitalize each word.
  • Line 25, throughout the MS, please be consistent with fish oil and change the control diet name to 20FO.
  • Here and elsewhere, report P uppercase and italic (P<0.05).
  • Throughout the MS, if there is no significant difference, no need to report P-value.
  • Please write the abstract more numerically about the important results. You can do it by adding their numbers in parentheses.

Introduction:

  • throughout the MS, please first mention the common name plus scientific name, and for the rest of the MS, just report the common name.
  • Line 37, please start with the problem related to farm fish at low temperatures. The 22 degrees is not strifeful for fish; they just do not grow well. Please delete “temperature stress” throughout the MS. Further, any response of fish to “temperature stress” which was mentioned in this MS is not relevant, like lines 37-41. Please update the MS from this point and delete them.
  • Line 42, Nile tilapia Oreochromis niloticus is…
  • Line 43, please add the annual production according to FAO 2020.
  • Line 47, 59 and elsewhere, when you say “low temperature” make sure you mention what temperature was. Only saying low temperature can be misleading.
  • Line 68-73, is it relevant to your study?
  • Line 82, please somewhere shortly report the results of your previous study, especially the growth rate.
  • Please update the introduction with recent works as many studies are available from the last two years, which were not included in this section.
  • Please mention the novelty of your work in the last paragraph of the introduction.

Material and methods

  • The statistical analysis should be updated with polynomial regression. It is possible that some factors have quadratic relations which the results, discussion, and conclusion can be totally different. Please check this paper to see how to report the Tables. Please update the MS with the change. https://www.sciencedirect.com/science/article/pii/S0044848619305861 This paper used both ANOVA and regression which is suited to your study as you can compare all 6 treatments if the P-value of ANOVA was significant.

Results

  • I will review the relative sections in the next version when you updated the MS with the new changes.
  • We can hardly see any numbers in results. Please add the number of some important results.
  • Tables 2,4, 6, and 8, I do not understand why you average the five treatments and compared with Control. What was the purpose?. Some important parameters such as Σ LC-PUFA n-3 is hugely different between 5AM (0.09) and 40AM (0.48). Averaging such different results can cause misleading. In Tables 1, 3, 5 and 7 please add a column for fish oil. Although you want to do the regression with 5 treatments, it is no harm to have fish oil there as well.
  • Please move Table 10 to the first part and change it to Table 1.
  • Please summarize the results, no need to report all details; readers can check all results from tables.

Discussion

  • Put the subheading for the discussion section like results. Also, keep a sequence in subheading for investigated factors, in M&M, result, and discussion.
    • As a general comment: please focus on fish as hips of references and studies are available, and no need to cite other vertebrates.
  • Some parts of the discussion are better updated with research in 2020 and 2021 as they refer to some old references. Please update the discussion with the latest studies as much as possible.
  • Although you wrote this section well, you can still improve it by answering these questions and annotated them to the discussion section. Why were these results observed? Discuss more possible reasons.
  • The conclusion needs to be revised and add more comprehensive concepts there.
  • I will review the MS when you updated it with a new analysis.

When revising your manuscript, please consider all issues mentioned in the comments carefully with clear outlines for every change made in response to the comments including suitable rebuttals for any comments you deem inappropriate. Please itemize your response to each comment, and highlight the revised at re-submission.

Best regards

Reviewer 2 Report

Manuscript ID: fishes-1346578

Title: “Dietary supplementation of Aurantiochytrium sp. meal changed body

fatty acid profile, gut and hepatopancreas morphophysiology of Nile tilapia

reared at suboptimal low temperature.” for Fishes.

1) General comments:

This MS describes effects of dietary supplementation of Aurantiochytrium sp. meal, rich in DHA, on weight gain and feed efficiency of Nile tilapia reared at suboptimal low temperature (22°C) to know an adequate winter diet for Nile tilapia aquaculture. This study is interesting to aquaculture managers and researchers. Results of this study would contribute to development of optimal rearing method for this species.

Comments:

L.89-91, “The fatty acid composition of the body was evaluated through regression analysis, presenting a significant linear response (p <0.05), where we observed an increasing or decreasing linear behavior of the fatty acids in response to the inclusion of Aurantiochytrium sp. (Table 1).”

Comment: I can hardly agree with the analysis results, especially in EPA and DPA (Table 1). There are no mistakes with these results?

L. 99-101 and several parts throughout MS, “When body fatty acid composition was compared between DHA sources, fish fed the diet containing 10 g kg-1 Aurantiochytrium sp. meal presented higher palmitic acid (PAL) 100 content and n-6 LC-PUFA than fish fed cod liver oil (CLO) (Table 2).”

Comment: Explain why the reason that authors compared fish fed cod liver oil with fish fed the diet containing “10 g kg-1” Aurantiochytrium sp. meal.

L. 145-155, “2.2. Viscerosomatic, hepatosomal index, and liver glycogen concentration”

Comment: Description of result in this section is so unfriendly for readers. Even if no significant difference was detected, it would be better off if the author would add new table or graph showing the results in this section.

I hope that my comments will help to improve this manuscript.

Reviewer 3 Report

Title

It should be reduced and improved.

Abstract

- Add the rationale of this study before the aim. See comment 2 in Introduction.

- Indicate which dose is the best and could be recommended.

Introduction

1) Authors split the introduction into several short paragraphs which it is not preferred. Some paragraphs have the same idea and should be combined. One idea = One paragraph. Please apply this throughout the whole manuscript.

2) The rationale of this study should be presented in a better way than the present one. Authors should indicate the shortage and limitation of their previous study and why they think in the current. Extension of the work could not be accepted as this could be a type of split for data. However, a strong rationale for the current study may overcome this problem. We want to know your hypothesis for this extension.

Results

- For histopathology, authors provided images for only two groups and ignored the remaining groups. They should provide all images for the four groups to compare them. For example, in Fig.2 only two groups were presented (3 images for control) and only one image for 4.0 g 100 g-1 group. You have to verify data presented in Tables 7-9 by providing us with all images for all groups. This is a mandatory request.

- In Fig.1: Why some images were highly basophilic, and the others were highly acidophilic? Is this a staining error or different color filter in microscope or image editing manipulation? If non of previous, authors should explain the reason for that and speculate about it in Discussion. In the legend at the name of the stain.

- Comparing Fig.1C with 1D, we could notice that number of intestinal folds are larger but the height is shorter in fish fed 2.0 g 100 g-1 of Aurantiochytrium sp. meal than the control fish. This was also shown in Table 7. Author should state that in Results and explain the reason in Discussion.

- Provide supported images for what you stated in lines 210-214. Higher magnification images are necessary to confirm that. Also it is so difficult to know the type of infiltrated cells using H and E so how you know lymphocytes and eosinophils?! Did you used specific set of markers or AB?! For this reason, we usually called them mononuclear cells. Line 212: what do you mean by “readiness”. Why you did not compare Cellular infiltration among different groups.

- Table 7: Add letters to indicate the significance between different groups and it would better to combine Table 6 and 7 if applicable. Why authors used polynomial regression analysis? It would be better and easier to use One way ANOVA.

- Fig.2: All image contained only hepatocytes, please select some images containing pancreatic tissue to evaluate pancreatic changes.

- Again, provide supported images for what you stated in lines 240-248. Higher magnification images are necessary to confirm that.

Line 145: Present these data in a table and indicate significance.

Line 188: What “.¹.” refers to? Really there are many “1” in this table which a little bit make reader confused.

Line 203: Which folds do you mean “the folds”? I know intestinal but make this clear at the beginning.

Line 205: “Figure 1).” Indicate which image A-D. Do the same with hepatopancreas images.

Line 152: Do not use different unit either write “2.0 g 100 g-1” or “20 g kg-1” but never both in the manuscript.

Discussion

- Summarized the main results in the first paragraph

Lines 365-367: “The rising levels of Aurantiochytrium sp. meal also resulted in a significant increase in the number of goblet cells; however, in fish fed the highest level of  Aurantiochytrium sp., there was a decrease in the number of goblet cells” This is really so confusing and could be rephrased using more meaningful words. Also explain why higher dose reduce goblet cells production. The highest dose show high body gain (your previous study) but reduce goblet cells (and so could decrease gut immunity). So what is recommended dose to be included in fish diet? You have to speculate about that in Discussion.

- Limitation of study and recommendation could be added before conclusion.

Materials and Methods

- The total number of fish and the number per each group are not clear.

- Number of histopathologically examined slides (per group) and sections (per slide) should be provided. Also indicate how these were examined (blindly or not).

- Indicate in statistical analysis how data were presented. Is T test appropriate for such statistics which contain more than two groups?

Line 445: How Aurantiochytrium sp. meal was mixed with fish diet?

Line 447: On what basis this dose was chosen (5, 10, 20, and 40 g kg-1)?  

Line 448: “had a similar DHA concentration “ Add reference(s).

Line 508: Write full names at first mention “VSI and HIS” Apply throughout the whole manuscript.

Line 510 and 511: What (1) and (2) refer to?

Line 514: Is this a revised version? See red letter “standardized”

Line 520: Delete ” were performed”

Line 551: Is it anterior third or proximal third? Previously you said anterior which is the most appropriate term.

Line 569: Change “picnosis” to “pyknosis”

Lines 571-547: Please rephrase this sentence using simple and clear words.